# Connecting Higher Education to the Labour Market: The Experience of Service Learning in a Portuguese University

**Sofia Castanheira Pais** [1,*] **, Teresa Silva Dias** [1] **and Deyse Benício** [2]

1 Centre for Research and Intervention in Education, Department of Education Sciences, University of Porto, 4200-135 Porto, Portugal; teresadias@fpce.up.pt
2 Faculty of Psychology and Education Sciences, University of Porto, 4200-135 Porto, Portugal; dgbenicio@gmail.com
* Correspondence: sofiapais@fpce.up.pt

**Abstract:** The mission of higher education institutions (HEI) includes fostering conditions that enable student participation, and a commitment to their professional success. Pedagogical strategies which combine learning that goes beyond the university and involves a service-learning (SL) course carried out in the context of a pedagogical innovation programme of the University of Porto were examined. This examination included interviews with teachers, a focus group discussion with students and the analysis of logbooks and final reports to realize the potential of SL for improving employability. The results put into perspective the weaknesses and potentialities of SL courses at HEI and show that when academic learning is integrated with community experience, students gain both personal/social and academic skills. They also develop leadership and communication skills and critical awareness on the one hand and time and resources management and the ability to adapt and respond to challenges of the real world on the other, all seen as beneficial for the transition into the labour market.

**Keywords:** service learning; higher education; labour market; learning; experience

## 1. Introduction

The university plays a nuclear role in achieving both economic growth and social progress. Beyond the instruction process and teaching–learning dynamics, university pedagogy also includes activities that facilitate engagement with various groups of stakeholders. Traditional university missions are thus being reconsidered to develop internal mechanisms that promote a high level of scientific production but that also allow training students with technical skills of excellence and personal, social and civic skills that enable them to respond to a society that needs creativity, solidarity and social justice [1,2]. It supposes the exercise of citizenship, not only attending to the search for scientific and technological knowledge but also in applying activities that are "oriented towards solving problems and responding to the demands of the community" (p. 138) [3]. Thus, the third mission of university implies a commitment to the communities and populations where they are located, identifying weaknesses or problems, perceiving existing resources and collaborating in the design and implementation of solutions that predispose to social change.

Assuming that the third mission is related to a change in universities' functions, its limits have always been a matter of debate amongst academics and society in general [4]. There are several ambiguous aspects surrounding the definition of the third mission. This is particularly true given Laredo's (2007) [5] categorization of the third mission based on the following three core functions: (a) educating the masses; (b) professional training and/or specialized research; and (c) academic training and basic research. According to this author, "universities do not structure their activities along these three distinct missions per se but articulate differently those missions depending on the functions they fulfil instead" (pp. 235–236) [6].

Therefore, one of the biggest challenges facing universities today tends to be the engagement and evaluation of the relationships with external stakeholders and communities. This constitutes one of the main reasons to define the third mission of the university around the term 'community engagement', considered "the set of activities consisting of community and a knowledge transfer function, through which the institution can demonstrate its relevance to the wider society and be held accountable' for its actions" (p. 237) [6] and which implies aiming to respond to problems in real time with the community of proximity. Nevertheless, and in line with other studies in this domain [7–9], the multifaceted nature of the third mission suggests moving from the periphery to the academic core of universities and accepting a set of unresolved challenges, both internally and externally.

Apart from the divergent ways of configuring the third mission, there appears to be consensus on the rejection of a neoliberal conception of university. Stating this, Menezes and colleagues (2018) [10] argue for an ecological and situated vision of the social responsibility of the university, advocating a real sense of social justice based on mutual respect and a strong commitment to the community [11]. Admitting this does not mean disregarding the expectations of the labour market but rather recognizing both needs and resources of the real world and having the university put more appropriate responses in academic, social and professional terms. Thus, this study aims to explore students' perceptions of the role of a service-learning programme in increasing employability prospects. It is expected to contribute to a deeper understanding of possible alternatives aiming to promote students' professional skills in higher education institutions.

## 2. From the University's Third Mission to the Labour Market—Spaces for Global and Citizen Education

The last two decades have been characterized by profound social and political changes in Europe, which have impacted the mission of universities [12,13]. This means that, while on the one hand, the 1990s brought up the instrumental side of higher education, emphasizing the international trade between institutions, on the other hand, the Bologna Process suggested the importance of a Europe of knowledge, highlighting new ways of structuring education activities and priorities [14]. Considering the emergence of this 'knowledge society', higher education institutions stressed their purpose in terms of citizenship. As Biesta (2009) [15] advised, higher education should not reduce adult education to 'learning for earning'. Rejecting the assumption that the economic dimension of higher education institutions is not relevant (or less so) compared to the social and intellectual dimension, institutions' contribution in diversifying what they offer should be considered.

The conflict between the logic of public service and the logic of the job market is reflected in the emerging 'educational market', which is substituting the 'educational state'. As at other academic levels, universities prepare students for life as active citizens in a democratic society, and this is an unavoidable challenge [13]. Assuming Dewey's inspiration, this tension could be solved based on both 'learning society' and 'lifelong learning' conceptions [16] as well as recognizing that citizenship is a nuclear ingredient of the mission of higher education institutions and so it should not be pressured from the outside nor limited to the curriculum. Moreover, it is the genesis of a wide range of higher education goals, making it clear that universities will always ensure universal values and cultural capital heritage [16].

Nevertheless, there are several aspects that interfere with the relationship between academia and the outside world. Technological development, demographics, globalization and climate change, among others, have reconfigured the world of work. Therefore, both the nature of the tasks carried out at work and the skills required in the labour market are pressuring classical education dynamics. In most cases, this implies searching for new approaches to lifelong learning and improving the quality of higher education institutions' strategies. SL constitutes one of these alternatives by combining experience within communities with academic purposes.

Beyond a traditional model of teaching and learning, SL integrates "meaningful community service with instruction and reflection to enrich the learning experience, teach civic responsibility and strengthen communities" (p. 3) [17]. In this sense, SL is an approach with the potential to develop competencies that are envisaged in comprehensive university education, such as civic–social, academic and professional competencies [18]. As a pedagogy, its aim is to achieve academic, civic and personal goals for students' learning through a close relationship between theory and practice [9]. In line with this perspective, Ash and Clayton (2009) [18] present a model that matches the components of SL—academic material, relevant service and critical reflection—into learning objectives that combine in a complicit and integrated way with the holistic education of students—academic learning, personal development and civic learning. Thus, SL intends to support and increase students' personal growth, as well as social, academic, professional and civic competences recognizing that learning also happens outside academia [19]. Considering Bringle and colleagues' model (2016) [20], academic learning, civic learning and personal growth are the three main domains of students' learning which are enhanced through SL. Academic learning is related to a deeper understanding of theories, academic concepts and research findings; civic learning is about gaining civic knowledge which is not exclusively related to the university curricula; and personal growth refers to developing personal values and attitudes which also might not be the focus of the course [21]. Thus, the focus of SL is not only on students' learning and development, but its aim is also to benefit communities.

## 3. The Experience of Service Learning in a Portuguese University

Higher education in Portugal underwent a reform in 1973, and it was from 1974 onwards that public higher education expanded. After democratization, which Pascueiro (2009, p. 33) [22] considers a "social and political value and/or as a process", entry into higher education became more accessible to new audiences. Later, particularly as a result of Bologna configurations, Portuguese higher institutions sought innovative strategies, new teaching–learning methodologies and adaptations to the new reality.

Considering that the practice of university social responsibility is a reality in European universities and also a political proposal, in Portuguese higher education, the promotion of the relationship of students with environments that depart from the formal context of education is, thus, an important goal [23]. This is the case of the University of Porto, in which investment in various projects involves exploring areas of knowledge in the academic community that can contribute to solving problems in local communities. Projects such as "University Social Responsibility in Europe" (EU-USR, 2019) (http://www.eu-usr.eu/), "Unibility—University Meets Social Responsibility" (2015) (http://www.eucen.eu/post/unibility) and "Promoting Social Responsibility of Students by Embedding Service Learning into Education Curricula" (2021) (https://www.engagestudents.eu/) reflect how valuable the participation of students in these practices might be.

As a pedagogical approach, SL has been growing in higher education, and authors such as Hurd (2006) [24], Butin [25] and Rego and colleagues (2020) [9] argue this is an interesting way to go because engaged students engage communities and engaged communities collaborate with local development and cooperate to add new research to universities. This highlights the assumption that since they are involved, students realize that learning extends to other spheres of their lives and that the concepts learnt can be applied outside the environment of the institution. Furthermore, SL is related to a conception of learning that is defined as a social process and that includes dialogue between the teacher, the student and the community.

Accordingly, Brigle and Hatcher (p. 1) [26] emphasize that only working on the acquisition of knowledge is not enough to develop intellectual skills. They advocate that, in addition, the education system should be expected to stimulate the desire and capacity for lifelong learning and activate the necessary skills in order to promote the democratic participation of students. Moreover, this is not exclusively related to academic development and success, but also includes a connection with professional integration. This seems to

be in line with the literature in this field since it reveals that applied learning such as SL allows for opportunities to reinforce academic concepts, which in turn facilitates retention and transfer of learning to social agencies and businesses or services [21].

The main goal of this study is to identify the perceptions that teachers and students have about the potential of service learning (SL) for improving employability from a course created within the scope of the faculty's pedagogical innovation programme. This is a curricular unit that was organized and coordinated by the Department of Education of the Faculty of Psychology and Educational Sciences, and its planning and implementation involved the collaboration of 14 teachers from different disciplinary areas (such as dentistry, engineering, economics, fine arts and architecture, among others) from the 14 faculties that belong to the University of Porto. It is an optional course, which can be attended by students of any subject area and different levels of study; it has a duration of one semester and confers credits inserted in the course plan.

More specifically, two objectives were defined to achieve the proposed global objective: (i) to understand the different contributions that a pedagogical approach based on SL has to students (based on students' and teachers' perspectives) and (ii) from the identified contributions, to ascertain which will have an impact on the professional context, i.e., which competencies students develop and whether they are somehow considered an added value when considering entering the labour market.

Regarding the objectives of the study, a qualitative research design was developed, with data collection from teachers and students at two moments: at a preliminary stage, to understand the expectations towards the course, and at the end of the course, aiming to understand how the process took place and the reflections and assessments of it by the target population.

## 4. Method

Assuming that the present study was based on a case study methodology, in the initial stage, individual semi-structured interviews were conducted with the lecturers and a focus group was held with the students; at the end of the course, the reports on reflection and assessment prepared by the lecturers and the logbooks (field notes and directed reflection at each moment of the development of the experiences in the SL course) prepared by the students were analysed.

The aim of adopting different data collection methods was to enable an analysis of the perspectives of a larger number of participants in this SL course, striving to follow up and monitor the implementation of this pedagogical innovation strategy.

The use of a qualitative methodology was considered to allow a situated perspective, based on a descriptive basis and constructed from the specific and unique perspective of the interlocutors participating in the research [27]. In this case, given the specificity of the course created and the analysis intended, we identified a perspective based on the process rather than the product [27]. Therefore, it focused on the meaning that both teachers and students give to the SL experience and, essentially, on the type of competencies that are developed and that will be of great value for the future professional environment.

### 4.1. Interviews

The use of interviews sought a more thorough understanding [28] of the teachers' conceptions of different pedagogical methodologies and the use of SL to connect to the communities and develop skills that complement the students' academic skills (personal and professional skills). The interview script was structured in five phases which allowed us to explore the teachers' opinions regarding (i) previous experiences within the scope of service learning; (ii) potentialities and weaknesses in the use of SL; (iii) contributions of SL in the development of academic, personal, social, professional and civic competencies; (iv) links with community partners; and (v) ingredients for the success of the course and expected results.

In this phase, eight interviews were carried out with teaching staff from different faculties who work with the implementation of SL in their courses (four male and four female teachers), as presented in Table 1. The following criteria were taken into account for the selection of the lecturers who were interviewed: (i) teaching experience; (ii) participation in management positions, namely pedagogic council, and integration in research centres; (iii) knowledge of course programmes based on the SL methodology; and (iv) experience in the implementation of course programmes based on the SL methodology.

**Table 1.** Information on the lecturers involved in the interviews.

| No. | Pseudonym | Age | Gender | Faculty | SL Course |
|-----|-----------|-----|--------|---------|-----------|
| 1 | A | 51 | Female | Faculty of Nutrition Sciences | Community Nutrition |
| 2 | B | 54 | Male | Faculty of Architecture | Porto, Territories and Invisibility Networks |
| 3 | C | 42 | Female | Faculty of Engineering | Project of Industrial Design |
| 4 | D | 40 | Female | Faculty of Dentistry | Community Intervention |
| 5 | E | 54 | Male | Faculty of Nutrition Sciences | Community Nutrition |
| 6 | F | 44 | Female | Faculty of Economics | Sociology of Organizations |
| 7 | G | 48 | Male | Faculty of Fine Arts | Final Project |
| 8 | H | 62 | Male | Faculty of Engineering | Project |

*4.2. Focus Groups*

Despite the possibility of defining eligibility criteria for inclusion in the focus group, the diversity that organizes it as a group is one of the differentiating elements of this method. It allows the framing of a group of people brought together by characteristics that define them as fundamental to be present in the same space, but, simultaneously, as individuals, they present different paths and identities in a space of social interaction. In this space, participants give their opinions, discuss and negotiate when opinions differ and build an overall perspective about the topic under discussion, which is more than the sum of the individual perspectives [28]. The option for the focus group with students was therefore intended to bring them closer to the natural interaction context found in the faculty and classroom contexts, while also allowing for a first approach and knowledge of the different students involved and their characteristics.

The focus group with the students used a similar script to that of the interviews with the teachers: conceptions about community service and previous experiences developed within the scope of SL; motivations for joining the course; contributions of SL in the development of academic, personal, social, professional and civic competencies; links with community partners; and difficulties and obstacles experienced. Students' involvement in SL programmes was the only criterion to consider in the organization of the focus group discussion. The students belong to different subject areas/courses as showed in Table 2, and four of them will start their professional careers in the following year as they are finishing their master's degrees. The focus group was conducted online using the Skype platform.

**Table 2.** Characteristics of the students who took part in the focus group.

| No. | Pseudonym | Age | Gender | Course | Degree |
|-----|-----------|-----|--------|--------|--------|
| 1 | A | 29 | Female | Education Sciences | 2nd Year, master's degree |
| 2 | B | 38 | Female | Education Sciences | 2nd Year, master's degree |
| 3 | C | 28 | Male | Psychology | 1st Year, master's degree |
| 4 | D | 22 | Male | Psychology | 2nd Year, master's degree |
| 5 | E | 28 | Female | Arts/Design | 1st Year, master's degree |
| 6 | F | 23 | Male | Architecture | 2nd Year, master's degree |

*4.3. Reports on Reflection*

The teachers' reports at the end of the SL course were also an instrument that was used to frame the analysis within the objectives of this study. The use of these reports was

intended to obtain a reflection on the evaluation of the process and the results obtained with the organization of this course of pedagogical innovation using SL.

The report is organized into six points: (i) summary of the programme; (ii) reflection on the results (formal and informal assessment, levels of student participation and involvement, monitoring and mentoring processes); (iii) reflection on the functioning of the course (dynamics implemented with the students, partners and the community); (iv) reflection on the process of the SL implemented in the context of higher education; (v) reflection on this course (what added value this experience has brought to the lecturer and students, what dynamics have shown the most potential and weaknesses); and (vi) space for suggestions for improvement regarding the course of the experience. The reports of three lecturers were analysed.

### 4.4. Logbooks

Within the scope of this study, the logbooks prepared by the students during the SL course were also used as instruments for data collection and analysis. The logbooks were organized into two dimensions—one of an instrumental or procedural nature (based on technical knowledge, planning and implementation of an intervention project) and another in terms of the expression of feelings (reflection focused on personal, social/relational and professional competencies). It should be noted that the logbook accompanies the whole SL process, involving the classroom phases and the stages and experiences carried out in context. In total, the logbooks of the 21 students who took part in the course were analysed.

### 4.5. Ethical Issues

The study was approved by the Ethics Committee of the Faculty of Psychology and Education Sciences of the University of Porto, and confidentiality and informed participation were ensured for all participants (students and teachers). The possibility of abandoning the study at any moment of discomfort was also safeguarded, as well as the possibility of not responding to or participating in specific moments of discussion.

### 4.6. Data Analysis

Data analysis was based on content analysis considering the dimensions set out when the interview scripts and focus groups were organized, but also the structure of the end-of-course reports and the logbooks as resources developed within the course, which envisaged the monitoring and follow-up of the course. Thus, we started from a referential framework based on theory and with predefined categories, maintaining an attitude of epistemological vigilance that would allow the emergence of new categories arising from the discourse (oral or written) of the participants in the study [29].

The analysis framed six dimensions—SL at the university; Operationalization of the SL course; SL weaknesses and potentialities; SL and the different interlocutors; Motivations and reflections—organized into 12 categories: service learning experience, space for the development of service learning; strategies used to involve and motivate partners in the field; project construction and implementation phases; monitoring and follow-up processes; advantages of the use of the methodology in the course; weaknesses/difficulties of the use of the methodology in the course; benefits of SL for students; benefits of SL for teachers; benefits of SL for communities; and motivations and reflections—which correspond to a total of 21 subcategories analysed.

As mentioned, based on the procedures underlying content analysis, the data collected from the different instruments underwent in-depth analysis in an attempt to "know [and interpret] what lies behind the words on which each of the participants in this study focuses" (p. 45) [30]. In a subsequent phase, the data were triangulated in order to define a table of categories capable of responding to the objective defined for this study—to understand what potential the use of the SL methodology has for increasing students' employability at the end of a study cycle/an academic degree, i.e., what personal and professional skills it allows students to develop and which constitute added value when they enter the labour

market. Table 3 presents the dimensions of analysis and the structure of categories that will lead the data analysis. After the categories and the indicators that define their specificity were established, all materials were coded using the excerpt as the unit of meaning [30].

**Table 3.** Dimensions and categories of analysis—potential of the SL to improve employability.

| Dimensions of Analysis | Categories | Subcategories |
|---|---|---|
| SL at the University | Space for the development of Service Learning | SL in the study cycle/degree |
| | | Course Unit |
| SL weaknesses and potentialities | Advantages of the use of the methodology in the course | |
| | Weaknesses/difficulties of the use of the methodology in the course | |
| SL and the different interlocutors | Benefits of SL for students | Academic |
| | | Personal |
| | | Professional |

In the data analysis presented below, for the purposes of protecting the identity of the participants, codes made up of initials and numbers are adopted to identify the discourse of those involved and the respective sources of data collection. We will use the following codes: interviews (Int, P1); focus groups (FG, D); logbooks (LB, C); final reports (FR, P1).

## 5. Results

The presentation of the results is organized according to the dimensions analysed in order to go into each one of them in greater depth: SL at the university (space for the development of Service Learning in the study cycle/degree; SL in the course unit); SL weaknesses and potentialities (advantages and weaknesses/difficulties of the use of the SL methodology); and SL and the different interlocutors (benefits of SL for students). Given the relevance of the last dimension in this study, we elaborate on each of its subcategories—the benefits for students at the academic level, the benefits for students at the personal level and the benefits for students at the professional level.

### 5.1. SL at the University (Space for the Development of Service Learning in the Study Cycle/Degree; SL in the Course Unit)

The experience of SL in the college context in its connection with one or more of the curricular units was considered very positive and fundamental by all participants in the study. Both teachers and students, in a concerted and unanimous manner, mentioned that the contact with the field (going to the practical contexts) which the SL promotes is fundamental for them because it allows them to put the operationalization of their theoretical learning in the college context into perspective. It "gives meaning" to the academic skills which students acquire throughout their training and enables them to understand how they can make proposals and develop their professional activity in the field. In various students' discourse, it can be perceived that this SL experience creates a balance between what they consider to be a strong focus of their courses on a theoretical framework, as shown in the following excerpt: "I hope to improve and complement my previous knowledge about community intervention that I had the opportunity to learn throughout the degree in education sciences, and mainly to put them into practice!" (LB, A).

Students and teachers also mention that the experiences of combining theory and practice require students to mobilize skills such as critical thinking, adaptability, and flexibility to articulate different knowledge and perspectives, as can be seen by what N says:

> "With this experience, I aim to strengthen my analytical and critical thinking skills, as well as gather more tools of theoretical foundation and practical mastery for my current work and for future initiatives and projects in which I may be

involved. (...) I think that group work will provide good spaces for discussion and formulation of strategies and understandings." (LB, N)

However, it is more than that; it also highlights that communities are part of the teaching and learning process, in the sense that the university develops and extends scientific knowledge based on the challenges and problems that arise in society, as P1 states in his final report:

> "On the one hand, this SL experience revealed to me several aspects of the theory that can be materialized and understood more thoroughly by students in their practical contexts. And, on the other hand, it showed that communities are absolutely part of our teaching and learning processes, in the sense that their real problems and strategies inform our way of thinking and problematizing community, social and education intervention." (FR, P1)

In fact, the use of SL experiences promotes close relationships between higher education institutions and institutions (public and/or private) of the community. This emphasizes the third mission of the university in social responsibility—to promote collaborative initiatives and dialogue with the communities to solve real problems: "very interesting to: promote closer relationships between the HEI and the community and highlight the importance of dialogue in finding solutions and developing strategies to improve community problems" (FR, P2).

The actions to be carried out within the framework of the SL may take on different characteristics according to the speciality areas involved, with teachers and students realizing and emphasizing the importance of organizing actions that meet the interests, needs and priorities of the communities rather than those generated in the context of the faculty:

> "...we participate in the project 'Smiling Paranhos', in which we do oral health promotion actions in schools and not only in schools, but also in other institutions that ask us for collaboration." (Int, P6).

Despite the admittedly positive character which the SL experiences can bring to the university context, in the written and verbal reflections made by students and teaching staff, it is possible to identify some negative aspects (which require deeper reflection when designing SL courses). These include the fact that the times and objectives of the course units are not always in line with the work which is felt to be necessary for the community. As mentioned by G, the impression was

> "...that the time we need to carry out the project may actually be longer than the time we have to complete this discipline. Because often, to be able to meet the objectives that are imposed by the discipline, by the people, by the faculty...we can reach the end and not feel satisfied that we still managed to do a good job because we needed to be more involved in the project." (FG, G)

*5.2. SL Weaknesses and Potentialities (Advantages and Weaknesses/Difficulties in the Use of the SL Methodology)*

When challenged to reflect on the potentialities of using SL as a pedagogical tool in a university context, and in the scope of this pedagogical innovation course, teachers and students are unanimous in reinforcing a work dynamic that allows academic content to be reconciled with practical examples and experiences, as referred to by P7:

> "I think that this year with this work methodology we managed to make a much more virtuous crossing between what the scientific literature has told us about those organizations and in these two major domains: on the one hand their structure and their functioning dynamics...and then bring this to practical cases." (Int, P7).

However, the focus is essentially on the empowerment that this dynamic promotes in students to work with the community, developing a critical and reflective spirit, but also creativity due to the diversity of contexts and people they come across. This encourages

them to take the other's perspective and, essentially, the ability to organize intervention proposals that correspond to all the intervening parties, students included. Thus, "there was great diversity in the proposals made in each group due to the characteristics of the contexts but also due to the background of training and interests of students" (FR, P2). Besides this fact, and due to the characteristics of this course that is structured so that different disciplinary areas can share and build knowledge, one of the potentialities highlighted is the diversity of perspectives and opinions that are generated in multidisciplinary working groups, as mentioned by M:

> "One of the potentialities of this curricular unit focuses on this existing synergy of knowledge, in which voices from different fields of thought are heard, an opportunity that I intend to make the most of. (...) I hope to develop various skills, particularly regarding teamwork, which is made up of various visions..." (LB, M)

In terms of weaknesses, teachers refer to three points for reflection: assessment, time management and the difficulty in reconciling common interests in a referential framework that corresponds to the characteristics of each subject area/student involved.

With regard to assessment, the teachers mention that this SL model requires an adaptation of thinking about the assessment methods normally used in an academic context. They are unanimous in considering that assessment should focus more on the process than on the results. Moreover, in this sense, they mention as a differentiating element the monitoring and follow-up process used in this SL course—the logbooks and mentoring—which allowed a more individualized follow-up with the students and a more global assessment perspective, where the level attributed at the end of the course is obtained from various assessment instruments: "I'd like to stress the use of the diary—which on the one hand allowed students to reflect on the process, its difficulties, and moments of gratification, and on the other hand for teachers to individualize the accompaniment of students" (FR, P2).

Time management is associated with the implementation of an intervention process in the community. From the formal point of view, it is important that students have time to approach and integrate the community contexts, time to assess the needs and existing resources and, finally, time to implement and assess the impact that the intervention had on that community.

*5.3. SL and the Different Interlocutors—Benefits of SL for Students*

The data presented below describe the opinions of the participants in the study (teachers and students) with regard to the benefits for students of involvement in SL courses. Based on the theoretical framework, benefits in three domains were heard: the academic domain, the personal domain and the professional domain. Although we are aware that the boundaries are blurred and that many of the benefits cross the three domains, we will try to highlight in each of them what stands out and, therefore, can be attributed to the integration in this component of contact with the communities which the SL proposes as a differentiating element.

a.   Academic

The benefits at the academic level were reinforced in the dimensions analysed above, with the coordination between theoretical knowledge and its operationalization in the contexts of practice standing out as the greatest benefit, meaning "a very simple modelling of these buildings to put on the 3D map to enrich the map. So this was a very interesting intervention that we had with the company, the company was very pleased with the work that the kids did..." (Int, P1).

From the reflection of teachers and students emerges the belief that qualification happens from action—one can only learn by doing and by doing, one makes intentional and consolidates the knowledge acquired. Thus, "since one of the objectives of the UC, Community Nutrition, is 'to develop skills in the design, implementation and evaluation

of community nutrition programmes', I considered that only field performance by our students would enable this objective to be achieved" (Int, P8).

Furthermore, contact with community institutions also implies a different perspective when designing and implementing community programmes, since these programmes must be based on the resolution of existing or emerging problems, but real problems. Meaning "to elaborate a project which, this time, would be about a real problem, in a community, also real. (...) I need to get involved and participate a lot and, in addition, I need to understand the need well and understand how I will be able to improve the life of a certain community, through the elaboration of a project" (LB, Y).

In general, working with other students in group work was also mentioned as a benefit at the academic level, as well as the enrichment of knowledge (peer mentoring) and the challenge of working in a team, with moments of negotiation and problem-solving when work perspectives seeming to be irreconcilable. "The diversity of students in the groups added some complexity to the work—combining visions and interests, understanding roles and possible courses of action—but it was also highly enriching of the students' experiences, clearly contributing to the learning opportunities of the course" (FR, P2).

b.   Personal

According to the teachers' perspective, the logbook was a platform that led each of the students involved to individual reflection and developed their critical thinking, particularly when the reflections suggested in the guide were associated with a more individual domain, related to their feelings and emotions towards the different situations experienced throughout the SL process (in the classroom context and in the relationship with institutions and people): "Their reflections on this made clear that most were highly engaged in the process and that these reflections were very formative" (FR, P3).

From the students' perspective, there were several competencies that they considered were developed. These are mostly relational skills but also greater civic awareness and the responsibility they have as citizens. In this sense, they reinforce the need they felt to adapt to the level of discourse and posture when working closely with people with different backgrounds and different interests. They emphasize the development of communication as one of the most important skills in relating to others: "a very important component is the soft skills component—the component that we are able to interact with afterwards, the clients with real life people who may not have the same technical knowledge. So it is important we are able to communicate in a language that everybody understands" (FG, P).

Additionally, according to the students, the fact that the experience in SL took them out of their comfort zone and "pushed" them towards new experiences contributed greatly to their overall development as a person, namely in terms of flexibility, organizational capacity and the ability to quickly adapt plans and redefine strategies, as can be seen in the following testimony:

> "I believe this experience also did me good, as I realized that I'm not the only one to plan much more than is possible to accomplish in a timely manner and that this is part of human nature. And it is necessary to accept this human tendency to want to 'harvest the fruits without giving the time to germinate' and become frustrated with it." (LB, GE)

Students go further when they reflect that the social realities they face make them develop a sense of mission and active citizenship. Being part of this SL course predisposes them to reflect on the impact that their actions may have on the interlocutors of the entities and the populations they serve, particularly when the objective is to empower people in the communities:

> "I think that college helps a lot to structure this type of activity, it gives another view that it is very possible to get involved in social projects. (...) and, that it is always good also to dedicate our time to dedicate our knowledge but going back to what João talked a bit about today of empowering people, of giving them the possibility of having a continuity in the project." (FG, A)

These factors make each student aware of the social responsibility they have in the society they are part of and make them aware of the importance of a community spirit: "to have felt this community spirit, let's say… It helped me a lot both in my academic expansion, as a person and also helped me to direct for example my professional life, my career. It helped me to expand other things. I look not only at my navel and see what I can do with what I have learnt or with what I have had. What I can do to reach more people" (FG, M).

More globally, we can say that an experience of SL predisposed them to the need to intervene more actively in society: "This whole process was an eye-opening experience for me because I was able to realize that we, as members of a community, have an active role, we have a voice, we can actually DO something. This project has shown me what is possible, that my community needs its members, and it has, in fact, motivated me to get more involved in work that benefits my community in the future" (LB, AM).

c.    Professional

At a professional level, teachers and students identify several benefits for students. First of all is the establishment of direct contact with communities and institutions/entities, which allows students to identify the area of intervention, get to know the work that is developed there and experience in a predefined and accompanied space/time, what could be a typical working day in their near future:

> "And, therefore, this contact allows students to understand particularly what is the production process of a certain article and so on. And, therefore, they get to know the manufacturing space of this company and they also get to know the administrative area, and they get to know the insertion of the company and that starts the initial bridges with the community where the company is installed." (Int, P7)

This contact and integration in communities and institutions both deepen after the first moments of observation and allows the student to begin to appropriate the philosophy and institutional culture, identifying the mission and vision of the entity, but also an appropriation of the institutional values. The training of this attentive and strategic look allows the student to identify and understand the dynamics of functioning, integration mechanisms, communication processes and leadership of communities and companies that are fundamental at the start of their professional career or at any time of career change: "students had the opportunity to contact with different interlocutors in the contexts of the practice (using video calls to conduct interviews; observe dynamics and activities; appropriate key documents to the context)" (FR, P3).

In a more concrete and operational way, this contact allows not only teamwork where a diversity of opinions and backgrounds coexist but also the development of leadership and communication skills. Thus, "those kinds of projects are also important because they foster a lot of skills, leadership skills, teamwork skills" (FG, P). Other skills are developed such as cooperation, working together and openness to different perspectives and ways of thinking, in a dynamic of creating shared value:

> "My group is very cooperative and respectful, they are people who are very open to new opinions and experiences, they are interested in whatever the project and discussion and bring interesting topics to it, so all this fosters a good atmosphere both in the group and for me. This is a good personal experience, at least now in the beginning where the objective and main interest is to build a good group base and support, good communication and interaction." (LB, IN)

On a more technical side, students also develop skills in planning the interventions and learn how to deal with challenges such as the management of time and resources and the choice of the most appropriate tools for the development of the projects: "helping a community, (…) being able to do a little bit for the environment, which I think is everyone's responsibility, (…). And the third has to do with, in my case, the career that I

intend to do that I would like to dedicate to this kind of project…because of the social and environmental issue" (FG, D).

In the context of practice, students are challenged daily to be flexible and to be able to deal with situations that require changes on the spot and the execution of the planned activities. Thus, "when we start listening to people and exchanging ideas with people, sometimes we see that what they need or what they want to do…or the need is completely different" (FG, M).

In conclusion, it can be stated that students unanimously recognize that this SL approach will enable them to develop not only professional skills but also personal and social skills, leaving a feeling of personal and professional fulfilment.

## 6. Discussion

This study is in line with the literature in the higher education field, revealing how important it is to assume the perspective of "educational market" [13] in training students to be prepared for life in society and, therefore, with a perspective of being active citizens in the communities where they live. Throughout the teachers' and students' discourse, the need for HEIs to assume a social mission is mirrored, framing the technical and intellectual skills (in relation to the economy and the labour market), but also developing a sense of social responsibility that students and teachers have to be involved in the development of the surrounding communities [10,15].

Furthermore, in agreement with previous studies [3,6], this study reinforces that the teaching strategies that flow between formal and non-formal spaces are valuable in enriching students' learning since they constitute effective ways of developing important competencies and skills for both professional training and work, such as problem-solving strategies, critical thinking, creativity and solidarity [1,2]. Thus, as the results indicate, SL seems to be one of those approaches that reinforce interpersonal and civic relationships in addition to academic learning skills [18].

The potential of SL, for both students and teachers, is evident. Teachers realize that communities are absolutely part of teaching and learning processes and seem to be in perfect harmony with what has been advocated in other previous studies when looking at the holistic education of students [18,20], with learning also taking place outside the academy [19] in what defines a USR view of the university from an ecological and situated perspective. On the other hand, teachers also state that this methodology is very enriching since it enables a closer approach and accompaniment of the students and their projects. As a pedagogical approach, and according to the definition suggested by Ryan (2012) [17], in this study, SL is seen as an inspiration for teachers to include different ways of doing and show that if involved in the community students might reinforce their learning through contact with the real problems.

From the students' perspective, the experience in SL pushed them towards new realities and promoted their organizational capacity and the ability to quickly adapt plans to solve problems, as well as providing a flexible response to the communities' demands. Thus, not only the ability to redefine strategies but also empathy are two dimensions students identified through this SL experience that affect their overall development as a person. This finding is then aligned with what was recommended by Brigle and Hatcher (1999) [26], when they state that the education system should prepare for professional integration and also promote democratic participation skills.

Moreover, teachers point to the main benefits of the SL experience, highlighting teamwork skills, communication skills, personality development, etc., in personal, social and professional domains. This was for many participants an experience of "giving meaning" to the academic skills which students acquire throughout their training as Santos Rego et al. (2020) [9] argued, understanding how they can make proposals and carry out their professional activity in the field and do it in a way that benefits communities from a perspective of shared social value [21].

The results show that when academic learning is integrated with community experience, students gain both social and academic skills while also developing critical awareness of the real world, all seen as beneficial to the transition into the labour market. Therefore, although this is a study located in a particular community, with a relatively small sample, it is believed that it can contribute to the reflection on possible alternatives to be welcomed by higher education institutions with a view to promoting students' professional skills.

One of the limitations of this study is the fact that other courses and areas of study are not represented here, from which one could perceive possible continuities or discontinuities useful for a more rigorous reading of SL as a strategy in higher education. Another weakness identified has to do with the dimensions of analysis mobilized for the context of this article. Furthermore, it would be suggested that future studies should monitor students' experiences over time in order to understand the practical effects of SL when entering the labour market.

This study contributes, on the one hand, to reinforcing literature in the field of in-service learning as a strategy with potential in the context of higher education and, on the other hand, to noting a set of arguments that allow considering the adoption of this approach in other specific contexts. The fact that a case study was developed allowed us to explore in-depth a set of characteristics of the in-service learning approach, as well as access concrete experiences, both from the point of view of teachers and from the point of view of students. Access to this information reveals the well-established dimensions of work from the SL, namely the positive effects inherent to the proximity between institutions and the community and the interest of the educational community to integrate SL into higher education institutions.

**Author Contributions:** Conceptualization, S.C.P. and D.B.; methodology, T.S.D.; formal analysis, S.C.P. and T.S.D.; investigation, D.B.; resources, S.C.P. and D.B.; data curation, T.S.D.; writing—original draft preparation, S.C.P. and T.S.D.; writing—review and editing, S.C.P. and T.S.D.; supervision, S.C.P. All authors have read and agreed to the published version of the manuscript.

**Funding:** This research received no external funding.

**Institutional Review Board Statement:** Not applicable.

**Informed Consent Statement:** Informed consent was obtained from all subjects involved in the study.

**Conflicts of Interest:** The authors declare no conflict of interest.

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
