# Peer review of "Connecting Higher Education to the Labour Market: The Experience of Service Learning in a Portuguese University"

_education, doi:10.3390/educsci12040259_

Round 1

Reviewer 1 Report

Review of manuscript “Connecting Higher Education to the Labour Market: The Experience of Service Learning in a Portuguese University”

The methodological approach of this paper should be framed under the terms of “case study” methodology. The authors have selected a single “case” (ie., Porto University) to study using multiple qualitative methods.

The objectives of the study should not be mentioned in the Method section (see 3 paragraphs in Method). The argument why this study should be conducted should come much earlier in the introduction. The reader does not know what this study is going to be about till the method section.

Essentially this study aims to examine students’ perceptions of the role of a service learning programme in increasing employability prospects. Since service learning is the core concept of this study, I strongly recommend that the discussion of service learning should come earlier instead of at the beginning of page 3. Although I understand the need for some grounding the context of the study in the introductory section, this section can be condensed and the definition of service learning and its role should come forward.

The authors should clearly mention what is the contribution of this study at the beginning of the introduction.

Although the authors mentioned a structured interview script, I am not certain of what kind of interviews were conducted. Please ground your choices of a specific method using well-established terms in educational methodology.

Perhaps the authors can leave an empty space before and after each interview excerpt to help the reader distinguish between the interpretation and the excerpts from the transcripts.

The discussion section needs some revamping. Be confident and state what is your contribution. As the section is now written, I could not distinguish what is the “new” contribution of this study- I only see the authors agreeing with previous works. What has this study added to our knowledge? How does this study differ from preceding ones? Is it a replication attempt only?

Explication of the practical implications for higher education is needed. Although I recognise that the authors vaguely mention about the implications, it is not clear what exactly they recommend. Be specific- what should higher education providers do in line with the present findings?

Limitations should be outlined- no single study is perfect. Engage in a discussion about selection and interpretation bias. The argument about generalisation from small samples is not a concern for case studies. Nevertheless, bring also forward the strengths of your methods, which are many.

*The term “authors” is used for my convenience.

Hope the comments could be of help!

Author Response

Agreeing with the reviewer's suggestion, authors decided to include the first three parts at the Methodology in the end of the of previous section. Moreover, the term “case study” was integrated in the first paragraph of the Methods section:

“Assuming that the present study was based on a case study methodology, in the initial stage, individual semi-structured interviews were conducted with the lecturers and a focus group was held with the students; at the end of the course the reports on reflection and assessment prepared by the lecturers and the logbooks (field notes and directed reflection at each moment of the development of the experiences in the SL course) prepared by the students were analysed.”

Considering that our argument was intentionally organized in three different parts included in the theoretical section, we are in line with the reviewer recommendation. It means we included in the end of the fourth paragraph the following sentence:

“Thus, this study aims to explore students’ perceptions of the role of a service learning programme in increasing employability prospects. It is expected to contribute to a deeper understanding of possible alternatives aiming to promote students’ professional skills in higher education institutions.”

Thank you for your advice. We included “semi-structured interview” to clarify the methodology technique:

“In the initial stage, individual semi-structured interviews were conducted with the lecturers and a focus group was held with the students".

Done; we agree that a space organization of the text should be improved.

We already point out that “Also in agreement with previous studies [3,6], this study reinforces that the teaching strategies that flow between formal and non-formal spaces are valuable in enriching students' learning, since they constitute effective ways of developing important competencies and skills for both professional training and work, such as problem-solving strategies, critical thinking, creativity and solidarity” (2nd paragraph of the Discussion). Nevertheless, some information was added in terms of contribution of the study, as well some aspects related with the limitations and potentials of it in the last part of the Discussion section.

“One of the limitations of this study refers to the fact that other courses and areas of study are not represented here, from which one could perceive possible continuities or discontinuities useful for a more rigorous reading of SL as a strategy in Higher Education. Another weakness identified has to do with the dimensions of analysis mobilized for the context of this article. Furthermore, it would be suggested that these studies need to be monitored over time, in order to understand the practical effects of SL when entering the labor market.

Even so, this study contributes, on the one hand, to reinforce literature in the field of in-service learning as a strategy with potential in the context of higher education and, on the other hand, to note a set of arguments that allow considering the adoption of this approach in other specific contexts. The fact that a case study was developed allowed us to explore in depth a set of characteristics of the in-service learning approach, as well as access concrete experiences, both from the point of view of teachers and from the point of view of students. Access to this information reveals the well-established dimensions of work from the SL, namely the positive effects inherent to the proximity between institutions and the community and the interest of the educational community to integrate SL into higher education institutions.”

Thank you very much for your valuable suggestions!

Reviewer 2 Report

COMMENTS:

 Comment 1

According to template file, please bold titles in the paper! Also, give numerical values on the titles and subtitles!

Comment 2

Prepare tables according to template file (form of tables is necessary to transform)!

FINAL OPINION:

Paper presents comparison between traditional model of teaching and service learning (SL) model. Also is possible to see, through the examination results, potentialities and weaknesses of the SL model.

Also, the goal of this study is to identify the perceptions that teachers and students have about the potential of service learning (SL) for improving employability from a course created within the scope of the faculty’s pedagogical innovation programme.

Abstract of this paper provide an adequate digest of the contents and introduction very detail describe previous research in this area. References clearly give an overview of the research. Authors uses the interview as a method of gathering the necessary information in this research. Sample used in this research is small.

This research is original!

ACCEPT AFTER MINOR REVISSION!

Author Response

Thank you for your comment and for your valuable feedback!

According to the template file, tables are now uniform.

Reviewer 3 Report

Text with an interesting and updated theme, focused on the relation between HEI and the labour market. The empirical dataset is valid. However, some information is lacking on the methodology for collecting these data. Regarding the results and the methodology of processing these data, which could enrich the analysis and answer some questions. The results of this paper need to be compared with the existing literature, which has highlighted the contribution of this paper.

Author Response

Agreeing with the reviewer's suggestion, authors decided to include  the term “case study” in the first paragraph of the Methods section (now reorganized), adding some more information about the methodology options:

“Assuming that the present study was based on a case study methodology, in the initial stage, individual semi-structured interviews were conducted with the lecturers and a focus group was held with the students; at the end of the course the reports on reflection and assessment prepared by the lecturers and the logbooks (field notes and directed reflection at each moment of the development of the experiences in the SL course) prepared by the students were analysed.”

We already point out that “Also in agreement with previous studies [3,6], this study reinforces that the teaching strategies that flow between formal and non-formal spaces are valuable in enriching students' learning, since they constitute effective ways of developing important competencies and skills for both professional training and work, such as problem-solving strategies, critical thinking, creativity and solidarity” (2nd paragraph of the Discussion). Nevertheless, some information was added in terms of contribution of the study, as well some aspects related with the limitations and potentials of it in the last part of the Discussion section.

“One of the limitations of this study refers to the fact that other courses and areas of study are not represented here, from which one could perceive possible continuities or discontinuities useful for a more rigorous reading of SL as a strategy in Higher Education. Another weakness identified has to do with the dimensions of analysis mobilized for the context of this article. Furthermore, it would be suggested that these studies need to be monitored over time, in order to understand the practical effects of SL when entering the labor market.

Even so, this study contributes, on the one hand, to reinforce literature in the field of in-service learning as a strategy with potential in the context of higher education and, on the other hand, to note a set of arguments that allow considering the adoption of this approach in other specific contexts. The fact that a case study was developed allowed us to explore in depth a set of characteristics of the in-service learning approach, as well as access concrete experiences, both from the point of view of teachers and from the point of view of students. Access to this information reveals the well-established dimensions of work from the SL, namely the positive effects inherent to the proximity between institutions and the community and the interest of the educational community to integrate SL into higher education institutions.”

Reviewer 4 Report

Point 1

Authors should justify the importance of this study based on strong literature review and other sources. The bibliography must be implemented.

Point 2

Learn more about paragraph 2

Point 3
Deepen the methodological information

Point 4

Furthermore the findings and their implications should be discussed in the broadest context possible and limitations of the work highlighted.
Future research directions may also be mentioned.

Point 5
Conclusion
Emphasize the importance of this study based on their significant findings.

Author Response

Authors reorganized the Method section and included the following sentence specifying that this is a case study and it integrated semi-structured interviews.

“Assuming that the present study was based on a case study methodology, in the initial stage, individual semi-structured interviews were conducted”.

Agreeing with the recommendation of the reviewer, authors included some more information in the last section, regarding both limitations and future directions of the study. Moreover, the importance of the study based on their significant was highlighted. The following two paragraphs were added:

“One of the limitations of this study refers to the fact that other courses and areas of study are not represented here, from which one could perceive possible continuities or discontinuities useful for a more rigorous reading of SL as a strategy in Higher Education. Another weakness identified has to do with the dimensions of analysis mobilized for the context of this article. Furthermore, it would be suggested that these studies need to be monitored over time, in order to understand the practical effects of SL when entering the labor market.

Even so, this study contributes, on the one hand, to reinforce literature in the field of in-service learning as a strategy with potential in the context of higher education and, on the other hand, to note a set of arguments that allow considering the adoption of this approach in other specific contexts. The fact that a case study was developed allowed us to explore in depth a set of characteristics of the in-service learning approach, as well as access concrete experiences, both from the point of view of teachers and from the point of view of students. Access to this information reveals the well-established dimensions of work from the SL, namely the positive effects inherent to the proximity between institutions and the community and the interest of the educational community to integrate SL into higher education institutions.”

Thank you very much for your valuable feedback!